

# Observations of mesospheric clouds in Latvia 1957-1983

Jānis Kauliņš

Institute of Astronomy, University of Latvia, Riga, LV-1004, Latvia

*Correspondence to*: Jānis Kauliņš (janis.kaulins@lu.lv)

**Abstract.** Until the 20th century In the 1980s, there were very few instruments for studying the upper layers of the atmosphere. Therefore, great importance was attached for observation of mesospheric clouds (MC) at an altitudes of 75-85 km., including also for amateur observations. With the development of space technologies, interest in them decreased in the second half of the 80s, but in the last decade it has grown again sharply. The reason for this is the observation of MC in places where they have not been seen before, but especially the evidence of the connection of MC appearance and parameters with the effects caused by climate change. Therefore, the study of the state and dynamics of the mesosphere and the analysis of long-term
processes has again become an urgent scientific task, which, taking into account the nature of the phenomenon and the environment under study, can be effectively carried out only within the framework of international cooperation, and MC observations from the ground are an important component of this task.

Long-term visual and photographic observations of MC were carried out in the Latvian branch of the All-Union Astronomical
and Geodesic Society. They started during the International Geophysical Year in 1957 and continued until 1983, that is, for 26 years. The observational materials accumulated in the archive of Latvian Astronomical society. The archive contains almost all observation logs of the mentioned period and more than 2,000 large-size photo negatives. Observations were mostly carried out according to a uniform, internationally recognized methodology, which has made it possible to obtain a unique, methodically comparable series of observations in terms of duration. The process of digitizing the observation logs has now
been completed; scanning of photographs is also planned. The article summarizes information about the content of the archive materials and the information obtained from it about the observation points and the photographic technique used in them, as well as an overview of the materials obtained during the observations and their content. Reasonable importance of continuing work and performing in-depth data processing.

## 1 Introduction

Mesosphere or so-called noctilucent clouds (MC) are one of those few natural phenomena where, even nowadays, even very simple visual or photographic observations can make a significant contribution to the understanding of the phenomenon itself and the processes related to it. First of all, it is related to the statistics of MC observability itself, which can be related to various processes both in the Earth's atmosphere and in outer space. After a some loss of interest in the second half of the 80s, in the last decade the interest in observing MC has increased dramatically. MC observed in places where they were not visible before

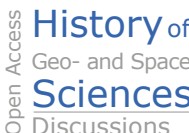

(Taylor et al., 2002; Nielsen et al., 2011; Hodorenko, 2020); there is also debatable evidence of the association of the appearance of MC with the effects caused by climate change (Lübken et al., 2018). Several authors (Kirkwood et al, Pertsev et al, 2014, Dalin et al, 2020) believe that such long-term trends have not been found, but these authors analyze NLC statistics and some physical parameters, while Lubken, among others, indicates this for the average height of the lower boundary of the NLC, which is clear downward trend, which has become significantly faster since the 1960s.

The study of the state and dynamics of the mesosphere and the analysis of long-term processes has therefore become an urgent scientific task (Fiedler et al., 2017), which, taking into account the nature of the phenomenon and the environment under study, can only be effectively carried out within the framework of international cooperation, and MC observations from the ground are an important component of this task.

Long-term visual and photographic observations of this natural phenomenon were made in the Latvian branch of the All-Union
Astronomical and Geodesic Society (AUAGS; since 1990 – Latvian Astronomical Society, LAS). They started during the International Geophysical Year (IGY) in 1957 and continued until 1983, i.e. 26 years. The observational materials accumulated in the LAB archive, which has been preserved as far as possible by the Museum of the University of Latvia (UL). The archive contains observation logs from the mentioned period and more than 2,000 large-size photo negatives. Some observation logs have unfortunately been lost because in the 1990s, it changed location and custodians several times. Photo negatives taken
after 1967 have not survived, too.

Observations were mostly carried out according to a uniform, internationally recognized methodology (Grishin, 1957), which has made it possible to obtain a series of observations that is unique in terms of duration. With some minor modifications, this methodology is also used by modern citizen science all over the world (Romeyko, 1990; Romeyko et al, 2003, NLCNET, 2022); the classification of morphological forms of MC has been supplemented (WMO, 2022).

At the same time, it should be noted that these data have only been processed to a very small extent and published to an even smaller extent. The processing took place mostly in accordance with concrete, narrowly specific tasks of the IGY and AUAGS. Most of the results are found only in special reports, which are not published; the few publications (e.g. E. Mūkin's article on the results of MC height determination) were in sources that are no longer available today and are known only from these same reports. However, the analysis of the series of obtained data and their details can provide important information about the
processes observed in the mesosphere in the previous years, its dynamics, and the possible connection with the meteorological conditions on earth. Considering the peculiarities of current observations, there is reason to think that it is also possible to obtain information about the nature of climate change during the observation period, which in this context has become quite important in the literature (Gadsden, Taylor, 1994; Oliveiro, Thomas, 2001). There are not many long lines of NLC observations (e.g., the one mentioned by (Romejko et al, 2003) – Moscow, 1962–2001. (Pertsev et al 2014) and (Dalin et al,
2020) refers to later years at the same place since 1990. Therefore adding another one will in any case improve our knowledge of the long-term dynamics of the NLC and the mesopause.



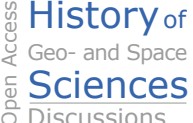

## 2 Systematic observations of mesospheric clouds in Latvia

### 2.1 Sources

The historical evaluation of MC observations made in Latvia is based on the following main groups of sources:

1. reports to the Central Council of AUAGS on the MC observation work carried out by the Riga (later – Latvia) branch (RB; LB),

2. field observation logs, in which the observers recorded the data required in the observation methodology directly and in real time: news about MM, meteorological data and notes on the photographs taken;

3. archive of original photo negatives.

In the LAS archive, there are two reports on the observation of MC in preparation for the IGY, as well as during the IGY (Dīriķis, 1957; Dīriķis, Bērziņš, 1958). They are copies of the reports sent to the Central Board of the AUAGS, and they are typewritten in Russian. The reports contain information about the observation points and their equipment, they describe the observation season and the main results obtained during it. The information attached below about observation points and observations during the IGY is obtained directly from these reports.

### 2.2 Observation points

In the territory of Latvia, regular MC observations within the framework of joint programs have taken place in four places: Baldone, Lielauce, Riga and Sigulda (Figure 1). The main information about observation points is added in Table 1. Observation log journals have survived from Riga, Baldone and Sigulda.

In Riga, the observations initially took place at 34 Gorkija (now Kr. Valdemāra) street, on the roof platform of a six-story
residential building. However, in 1958, shortly after the beginning of the observation season, due to technical reasons, the observation site was moved to the Riga Palace, where the cultural center of the ideological children's organization "Pioneers" was located at that time. Observations in one of the palace towers began there in the middle of July 1958 and continued until at least 1960. In the later years, it was observed in Riga from the territory of the Astronomical Observatory of the UL (now the Satellite laser ranging station of the Institute of Astronomy UL; see figure 3). At the observation point there was a photo
camera АФА-ИМ, binoculars, a wide-angle theodolite and a radio receiver for receiving accurate time signals. During the IGY, the work of the point was led by E. Grasbergs, a student at the UL. After 1966, systematically, but after 1969 – no sightings have taken place in Riga and all active activity takes place only in Sigulda.

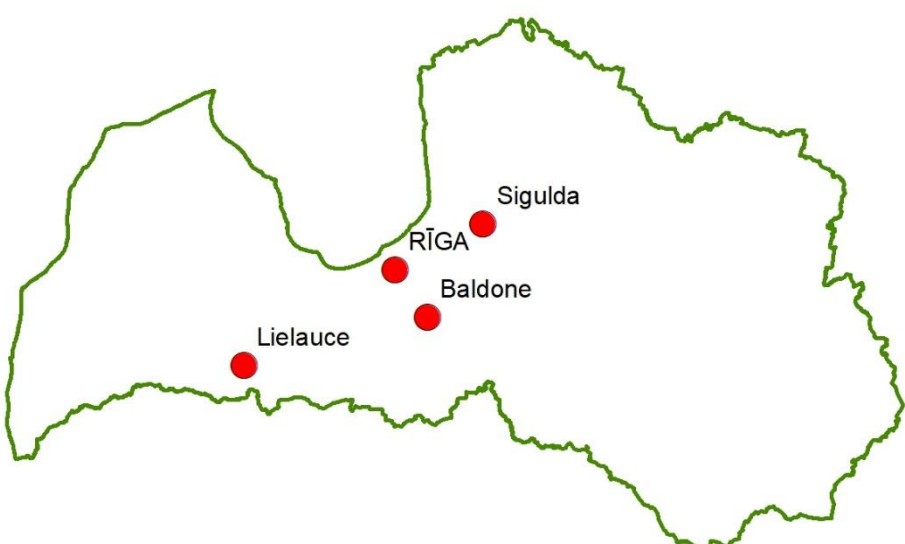

**Fig. 1 Location of MC regular observation points in Latvia during the IGY (figure in color)**

In Sigulda, the observation point was located in the territory that used to be at the disposal of AUAGS RB. This territory was granted to the association for use by the decision of the local administration in 1956, and already that year a special pavilion for MC observation was built there (figure 2). Observations began in 1957 and took place there until 1983. The observation point was well equipped. It housed aerial photo cameras АФА-ИМ and НАФА-3с on a masonry pillar with a deep, stable foundation. Other instruments include a light-capable wide-angle theodolite, a geodetic theodolite and binoculars. The needs

of the weather service were served by a radio receiver for receiving precise time signals, marine chronometers and an aviation chronometer, and in the 80s also a quartz clock. During all observation periods, the activity of the point was managed by Matīss Dīriķis, a scientific associate of AO UL.



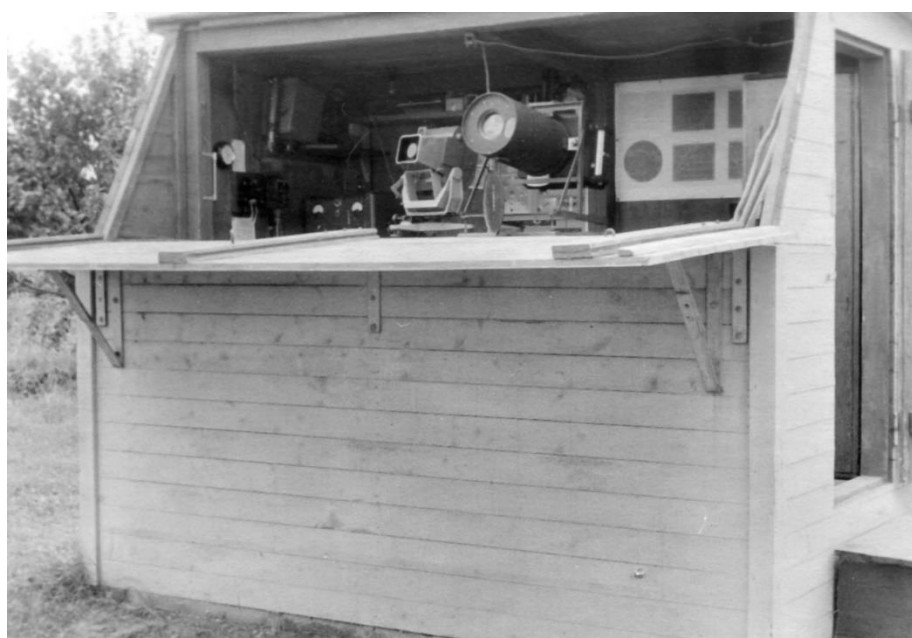

**Fig. 2 MC observation pavilion at AUAGS RB observatory in Sigulda. Photo by M. Dīriķis, 1958.**

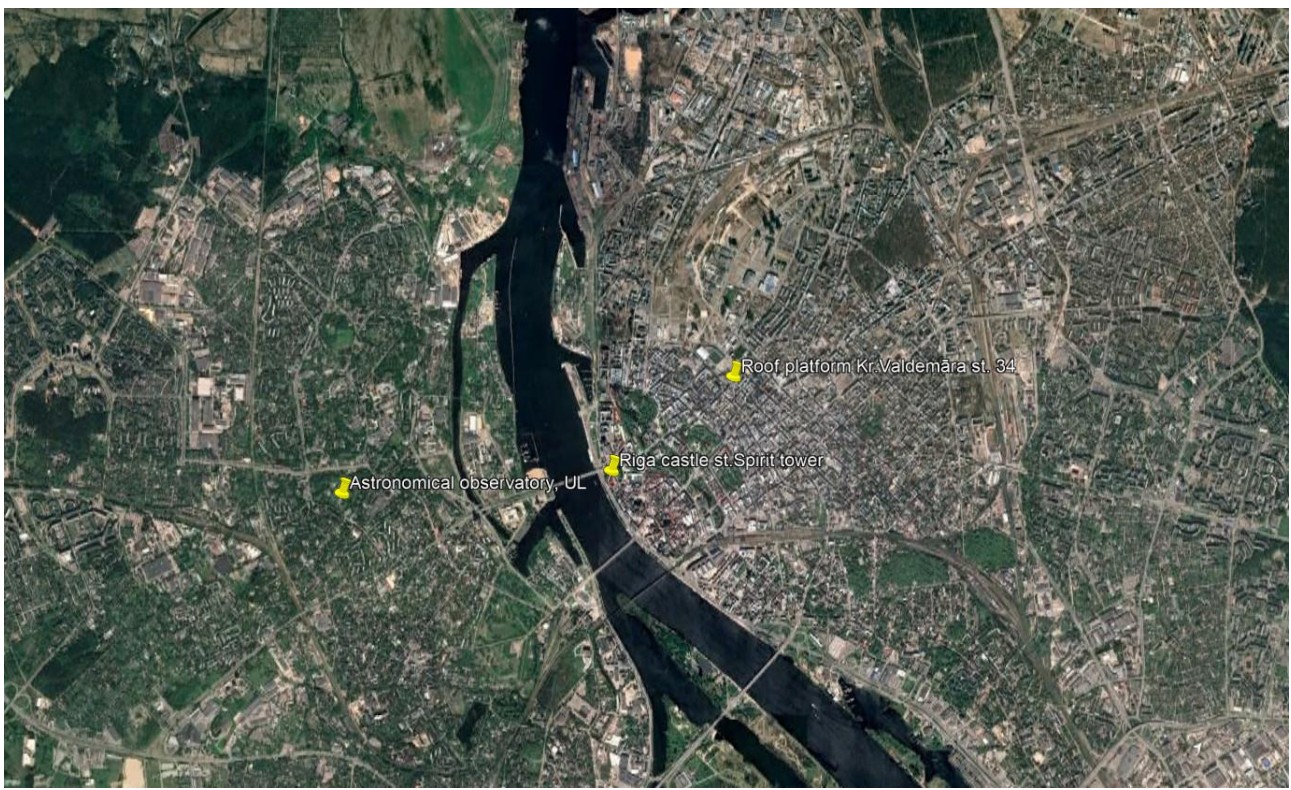


**Fig. 3 Places where MC observations took place in Riga. Background map: Geospatial information agency, Latvia.**

In the center of the picture are wide film cameras АФА-ИМ and НАФА-3c. The thermometer and barometer used in the observations can be seen on the left side. The opening of the pavilion is oriented towards the north. The pavilion, also called the "Cloud Hut", remained until the end of the observatory's existence in 1993, but since 1988 it was no longer usable due to the deterioration of the wooden structures.

**Table 1. Main characteristics of MC observation points**

| Location | Coordinates in source | Real coordinates by ortophotos | AMSL, m | Address | Observation period |
|---|---|---|---|---|---|
| Baldone | 56°46' N, 24°24' E | The exact place is unknown | ≈75 (terrain) | Astrophysics laboratory of Academy of Science, Baldone, Riekstukalns | 1958, 1959 |
| Lielauce | 56°31'25" N, 22°54'08" E | 56°31'22" N, 22°53'57" E (visual) 56°31'29" N, 22°54'05" E (photo) | 115 (building) 100 (terrain) | Geodetic training ground of the Latvian Academy of Agriculture, Lielauce | 1957, 1958 |
| Riga | 56°57'33" N, 24°07'14" E | 56°57'32" N, 24°07'07" E | ≈28 (building) | Gorkija (now Kr.Valdemāra) street 34, roof platform | 1957 – 1958 |
| Riga | 56°57'09" N, 24°06'04" E | 56°57'03" N, 24°06'04" E | ≈29 (building) | Pils square 3, Riga palace St.Spirit tower | 1958 – 1960 |
| Riga | No data | 56°56'54" N, 24°03'33" E | 8 (terrain) | Kandavas street 2, In the territory of the Astronomical Observatory of UL | 1965 – 1969 |
| Sigulda | 57°09'38" N, 24°51'09" E | 57°09'39" N, 24°50'58" E | 100,5 (pillar) | Lāčplēša street 18, AUAGS RB (LB since 1961) People's Observatory | 1957 – 1983 |

The Lielauce observation point was located on the territory of the Latvian Agricultural Academy (LAA; now – University of Biosciences and technologies) geodetic training ground. The observation area was on the S bank of the lake, where the horizon was completely clear, but there was no pavilion. Therefore, visual observations were made from the balcony of the Lielauce Castle (where the LAA training base was located at that time), but in the event of the appearance of MM, it was possible to get to the 250 m distant observation area within a few minutes, where it was possible to quickly install the АФА-ИМ camera. For this purpose, a brick post built for the geodesy polygon was used. Looking from the building, the horizon on the NW side was partially covered by the trees of the park, so a full-fledged observation point could not be installed there. The observers also had the geodetic instruments of the LLA training base at their disposal. The work at the observation point in 1957 and 1958 was led by LLA geodesy lecturer L. Ozols. There are no reports of sightings in this place after 1958. Observation logs have not been preserved either, but in the Riga logs we find notes that MC were observed in Lielauce.





Observations in Baldone took place in the territory of the Astrophysics Laboratory of the Academy of Sciences of the Latvian SSR (now – Baldone observatory of Institute of Astronomy, UL). An observation pavilion should have been built there already in 1957, but it has not been done. Therefore, in 1958, observations at this place were made only visually, and only during the period when observations in Riga were not possible. D. Melnāre was responsible for the observations and they were also carried out. In 1959, full-fledged (i.e. also photographic) observations took place there, but there is no news about later years. It was also not possible to identify the exact location of the observations, therefore its coordinates are known only with the accuracy mentioned in the reports up to a arcminute.

**2.3 MC observation logs and their digitization**

In the archive of the LU museum, it was possible to find journals that testified to MM's observation work at observation points in Riga, Baldone and Sigulda. Riga observation logs were from 1957 to 1969, missing 1962. The records of the Sigulda observation point refer to the period from 1958 to 1983, missing 1957, 1968. and for 1970-1972. An observation point in Baldone is represented by one – 1959. In 1958, the log journal was started in Baldone, but the second half of the season was continued in Riga. Observation logs from Lielauce point have not been found.

The observation logs were digitized in the form of MS Excel tables; this work was done by the author of the article and volunteer assistants, students J. Stepanova and J. Biķis. In order to avoid arbitrary interpretations, a small instruction was prepared for the volunteers - explanations about the work to be done. A separate Excel file with 4 sheets was created for each year. Page 1 lists the data entrant, provides key information about the observation point, and provides explanations to help you understand the material that follows. Page 2 contains a list of observation nights and observers, information about clocks and their correction, as well as pre-processing data on the fact of MC observation and the number of photographs taken. Page 3 contains a full record of night observations, including the fact of MC detection, brightness, information on morphological forms, as well as meteorological parameters. Page 4 contains information about the photos taken.

Sometimes logs contain entries (notes) that are too long to fit comfortably in an Excel sheet cell. Therefore, only a reference to such notes has been placed in the relevant place, but they can be found in the appendix - in a text file. The digitized observation logs and the text appendices are available in the LU e-resources repository at https://dspace.lu.lv/dspace/handle/7/61099.

Almost all observation entries in logs have been made in accordance with the requirements of the methodology (example in Figure 4). For reasons that are difficult to understand, the requirement to take notes of observations only in pencil was introduced of the time of IGY and later remained. Sometimes, probably due to the lack of experience of the observers, the parameters of the observed MMs (brightness, morphology) have not been fixed.

Initially (1957, partially until 1960) the notes were made in Russian. In these cases, the digitized version is a Latvian translation made by the author (the author is fluent in Russian). In some places, the use of the Cyrillic alphabet has caused difficulties for the observers, who in most cases have been Latvians: the letters "D" and "Д" are represented in the handwriting in the same way, but correspond to different degrees of covering of the twilight segment (respectively, D and E); the same applies to "B"



and "Б". The latter clearly corresponds to the letter "B" of the Latin alphabet, but the first, which is written like the Latin "B", is the 3rd letter of the alphabet in Cyrillic and represents the grade C. The records of the Riga observation point for 1967-1969 are very incomplete. In 1967, it is obvious that they were carried out practically only on those nights when MC were observed. Regarding 1968 and 1969, it cannot be stated with certainty. In the first years, meteorological conditions were not recorded for some of the observations in Riga.

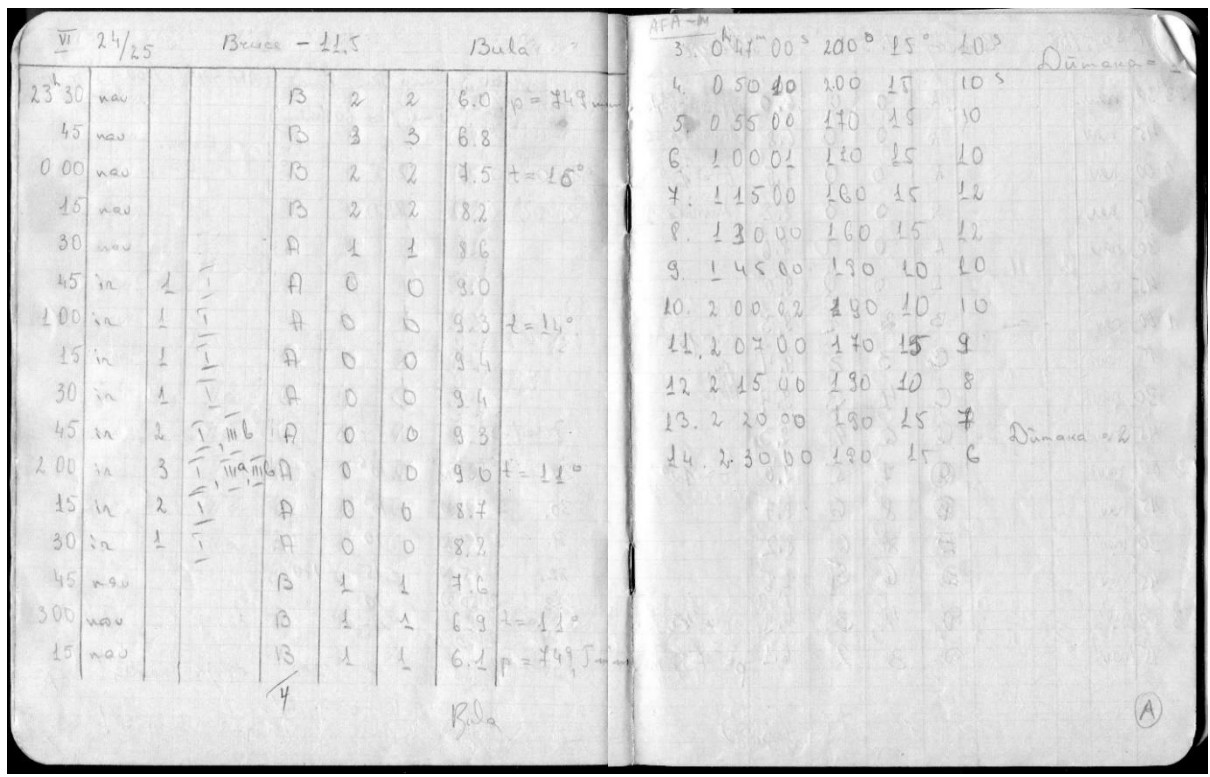


**Fig. 4 Opening of the observation log with MC and meteorological observation notes and information about the photos taken. 1976 year. Image of the author.**

**2.4 Visual observations**

Systematic observations of MC, which took place in the territory of Latvia in the period from 1957 to 1983, were mostly
carried out in accordance with the methodology described in (Grishin, 1957) and (Bronshtein, Grishin, 1970), observing some of the above-mentioned nuances and later additions.

Observations during the IGY season began in March and lasted until the end of October, and in the later period - on average from the beginning of June to August. The results of the observations were recorded in a certain sample journal, which was arranged separately for each year (exceptions: Riga, where the 1966-1969 journals are in one volume, also Sigulda, 1981 and
165 1982).

In visual observations, the visibility and typology of MC was recorded at the beginning of every quarter of an hour, with the time recorded to the minute. Clock correction, if used, was noted at the beginning of the observation session. The recording of the moments of time in Sigulda was ensured by the marine chronometer made by the company "J.Bruce&Sons" around 1900; later also a similar "Thomas Mercer" instrument whose correction was marked by accurate time signals from dedicated AM

band time signal transmitters. Sometimes the aviation chronometer AVRM was used in parallel. Since 1981, a "household" type quartz clock was used, and in 1983 – a quartz clock built by V. Gedrovics, an amateur astronomer and electronics engineer. At the other observation points, radio time signals were listened to, after which the hand (sports) stopwatch was started at the beginning of the first full hour of observation. In observations at Kandavas Street 2, the observatory's time service quartz clock was also sometimes used.

Air temperature and pressure were recorded only in observations in Sigulda, starting from 1964. Sometimes, if the weather and its forecast in the evening have been "hopeless" – E/10/10, the observers have not been on duty all night, but only noted such conditions for some periods at the beginning or even immediately and have not continued the observations. In Riga, meteorological parameters have not been fixed at all for some years. Observers have sometimes made logical errors in cloudiness estimation. For example, the total cloudiness is rated with a lower reading than the low clouds, or at cloudiness

indicators of 10/10, the cover of the twilight segment is rated as D. However, such errors are rare. In the early observations in Riga, smoke from factory chimneys was quite often mentioned as a factor disturbing the observations: north of the observation sites in the center of Riga is the industrial district of the Port of Riga. Here is an interesting conclusion about atmospheric industrial pollution and air quality – for many years (at least since the mid-1980s; author's observation) nothing similar has been observed in the city.

**2.5 Photographic observations**

**2.5.1 Photo archive of MC observations**

Almost all images have a frame size of 180x130 mm; For 84 images, the format is 180x240 mm. There are a total of 2106 photo negatives in the archive. Of them, 271 were admitted in Riga from 1958 to 1964, 1482 were admitted in Sigulda from 1961 to 1967, 146 were admitted in Baldone in 1959 and 8 in Vecauce in 1958. 106 pictures make up 53 pairs taken

synchronously in Sigulda-Riga and Sigulda-Vecauce. Most of the images are placed in binders and organized by series of observations. Each photographic negative is assigned a unique common identification number. The photo negatives are placed one by one in an envelope made of a folded sheet of soft paper (Figure 5) and are annotated with a description of the shooting conditions. All negatives are perfectly preserved. 186 mm wide aerial film was used for shooting.

Annotation for each photo includes:

observation point code (initial letter of the name),

personnel number (usually in the format XX-YYY, where XX is the last 2 digits of the year, and YYY is the serial number, which starts from 1 every year),

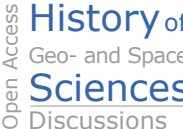
year, date and time of enrollment,

duration of exposure,

camera direction (azimuth, height),

the initials of the observers who made the frame.

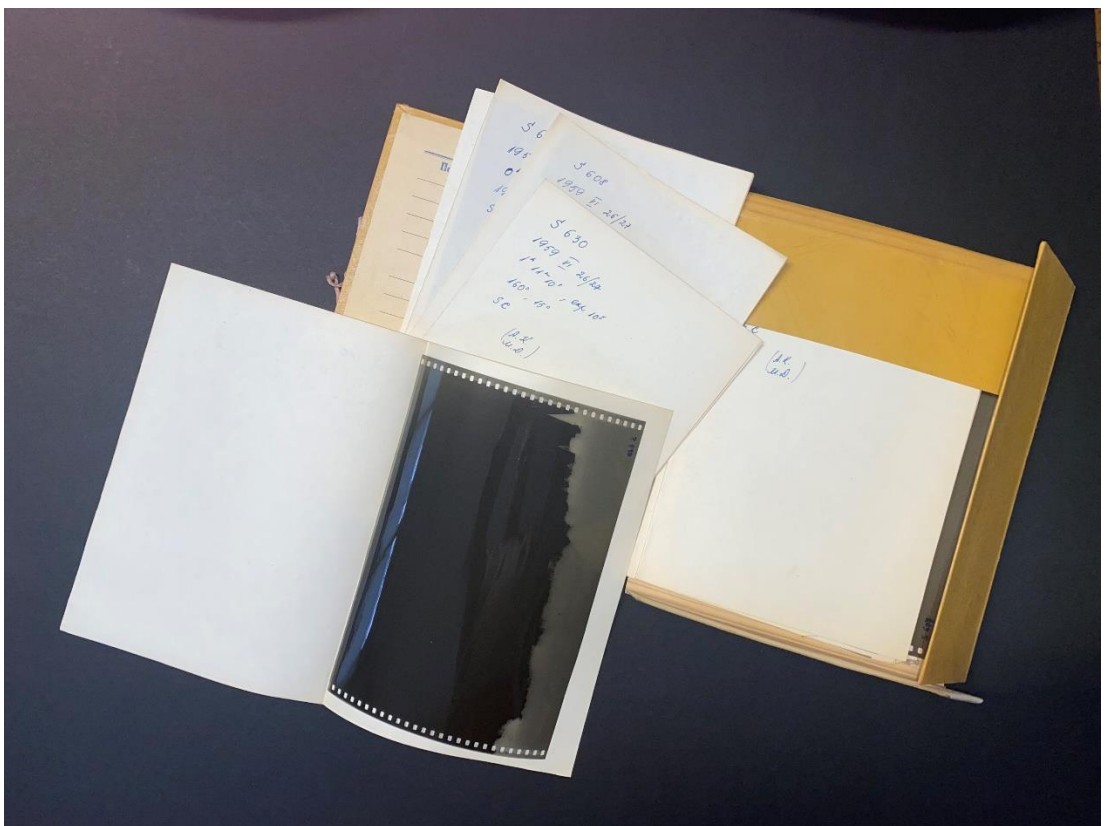

**Fig. 5 MC 180x130 mm photo negative archive folder. Author's photo. (figure in color)**

A semi-transparent paper template made during earlier processing is attached to some of the images, indicating the positions

of the brightest stars in the frame and other marks.

It is planned to digitize these images at the Institute of Astronomy of the UL. It is intended to be done with a professional high

performance scanner Epson 11000XL. Such scanners were used to digitize the 24x24 cm glass negative library of the 120/80

cm Schmidt telescope of the Baldone Observatory (Eglītis, Eglīte, 2016). An auxiliary device for accurate positioning of

flexible photo negatives on the scanner glass was made at the Laser Location Station of Institute of Astronomy, UL.

Photography was only performed if MMs were detected visually; the exception was some trial expositions.



### 2.5.2. Equipment

For serious photographic observations of MC in the USSR, it was recommended to use aerial cameras with a frame size of at least 180×130 mm and a normal lens (focal length equal to the diagonal of the frame). The large size frame was chosen to allow for image processing with photogrammetry methods. The relevant cameras of the local branch of the AUAGS (usually

they were decommissioned military aerial cameras) and special, ultra-fine-grained photographic materials could be obtained from the Central Council of the AUAGS. It was not acessible to individual, "unorganized" observers.

Aerial photo camera АФА-ИМ was used for MC photography in Latvia (figure 6; Abramov, 2022a). It was developed already in 1938 for the needs of the military department of the USSR. The camera used a four-lens anastigmatic lens Industar-51 with f=210 mm and aperture 1:4.5; the angular size of the frame was 46×30 degrees. Always used the largest aperture of the lens.

A semi-automatic aerial photo camera НАФА-3с with a frame size of 180x240 mm was also used in Sigulda for a short time. The diagonal of the frame was 27 degrees of arc. The built-in lens was Industar-52 (f=500 mm, aperture 1:5) (Abramov, 2022b). Not to be confused with the lens of the same name, which was widely used in the USSR for amateur small format cameras; it is a completely different device. This technically more developed camera was only used for a short time. The performance of the АФА-ИМ camera was recognized as sufficient and did not require special training, while the use of the

НАФА-3с required certain skills. In later years, 35 mm format cameras FED-3 and Viliya were also used in systematic observations. It should be noted that frames obtained with an amateur camera on 35 mm film cannot be used for photogrammetric analysis; they are also not found in archives.

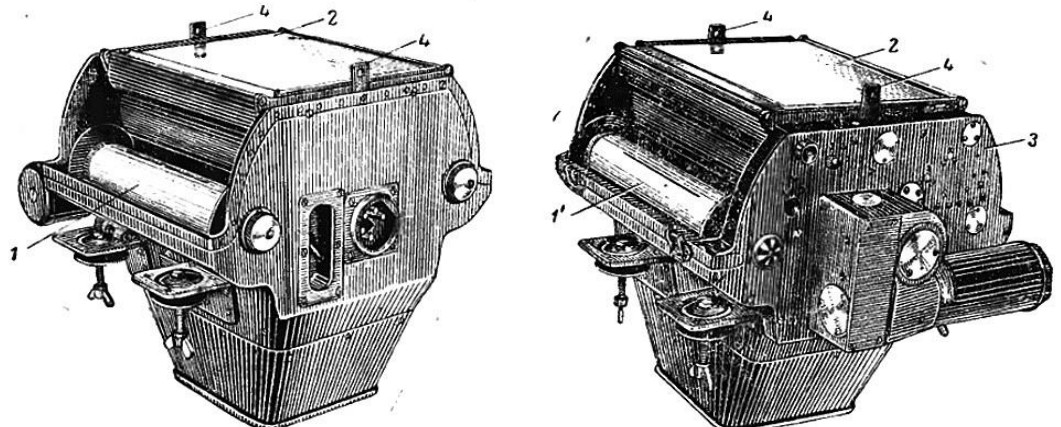

**Fig. 6 Aerial photo camera АФА-ИМ with the cassette cover removed**

1 – receiving coil, 1' – feeding coil, 2 – pressure glass, 3 – removable side part of the camera, 4 – cassette cover latches.

Photographic observations took place in Riga, Sigulda and Lielauce. Panchromatic aerial photo film with optical density D = 0.85, contrast ratio γ = 1.7 – 2.1 and sensitivity 1000 – 1900 units of the state standard (ГОСТ) was used as photographic material. This special aviation film standard of the USSR is not directly comparable to the well-known ISO, ASA and DIN standards or soviet ГОСТ for civil industry products. Similar to ISO, it is in the arithmetic progression, but the conversion

factor is unknown. The exposure duration was determined according to a special, experimentally obtained table (Table 2). The

exposure depended on the depth of the Sun below the horizon, which made it possible to calibrate the brightness of the MC

depending on the degree of blackness of the exposed photo film, eliminating the subjective factor.

**Table 2 Dependence of exposure times on sun depth**

| $-h_s$, arc deg | duration, s | $-h_s$, arc deg | duration, s | $-h_s$, arc deg | duration, s | $-h_s$, arc deg | duration, s |
|---|---|---|---|---|---|---|---|
| 6.0-6.6 | 1 | 8.2-8.4 | 6 | 10.3-10.8 | 20 | 12.4-12.6 | 60 |
| 6.7-7.2 | 2 | 8.5-8.7 | 8 | 10.9-11.0 | 25 | 12.7-12.9 | 80 |
| 7.3-7.5 | 3 | 8.8-9.3 | 10 | 11.1-11.4 | 30 | 13.0-13.2 | 100 |
| 7.6-7.8 | 4 | 9.4-9.6 | 12 | 11.5-11.9 | 40 | >13.2 | 120 |
| 7.9-8.1 | 5 | 9.7-10.2 | 15 | 12.0-12.3 | 50 | | |

In order to ensure synchronous photography of the same MC field, exposures had to be carried out according to a specially

determined time-direction table in the IGY observation programs in 1957-1959 (see Table 3). If the MC is weak, without

interesting shapes and significant details, then it is permissible to shoot only every full tenth minute. Observers continued to

adhere to this scheme later, as long as there were systematic observations from at least two points, approximately until 1966.

In later years, there was only a recommendation to do it at the beginning of a minute when choosing the starting moment of

photography. The start of the exposure was determined by hearing, listening to the exact time signal and waiting for the full

minute-long beep. For this purpose, DIZ (Germany, 4.525 MHz) special time signal broadcasting station was most often used;

less often also some others. Observations were often hampered by the poor audibility of these signals in household radio

receivers (then a stopwatch was visually used), but this did not interfere in Sigulda, where there was a precisely tuned military-

type radio receiver with variable selectivity and an external antenna.

**Table 3 Mesospheric cloud synchronous capture program.**

| Azimuth, degrees | 150 | Optionally | 180 | 210 | Optionally | 240 |
|---|---|---|---|---|---|---|
| Time, one minute after the start of each full hour | 00 | 10 | 20 | 30 | 40 | 50 |
| | 02 | 12 | 22 | 32 | 42 | 52 |
| | 04 | 14 | 24 | 34 | 44 | 54 |
| | 06 | 16 | 26 | 36 | 46 | 56 |

Briefly about the main results.

A brief overview of MC observations in Latvia in the period from 1957 to 1983, which can be found in the observation journals,

is attached in Tables 4 and 5. It can be seen that the mid-1960s and 1970s stand out with a particularly large number of MC

observations, but in 1957 there was only one and a dubious observation.



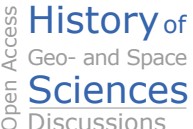

## 3. Discussion

In Latvia, a very large amount of MC observation material has been accumulated, the processing of which was carried out at the time did not nearly exhaust the potential hidden in it. A long series of observations about the fact of the appearance of MM, detailed (photographically) recorded topography and morphology of MC can provide a lot of new information about the long-term dynamics of MC and its connection with climate change processes taking place in the troposphere. Specialists from Institute of Astronomy, UL, attracting students and volunteers, will continue to perform statistical processing of digitized

information. Special attention should be paid to the rich photo archive, which can be digitized and processed in several directions. MC height measurements should be taken first if possible. Evaluating the results compared to those obtained, for example, in (Lübken et al., 2018), it is possible to confirm the assumption that they can serve as a large-scale indicator of climate change. Studies of MC movement and morphology can provide valuable information about the dynamics of the upper atmosphere and its long-term changes, the relationship with the solar activity cycle and other factors, including the already

mentioned climate changes. Among other things, an a priori statistical evaluation of the observations shows an apparent inverse relationship with the solar activity (Wolf number), but it will be clarified and substantiated in the further course of data processing.

**Table 4 Review of systematic observations of MC in Latvia: Sigulda**

| Year | 1958 | 1959 | 1960 | 1961 | 1962 | 1963 | 1964 | 1965 |
|---|---|---|---|---|---|---|---|---|
| Observed, nights | 153 | 76 | 52 | 64 | 55 | 48 | 83 | 81 |
| Certain MC sightings, times | 4 | 8 | 3 | 8 | 3 | 16 | 15 | 15 |
| Certain MC sightings, hours | 4.25 | 21.25 | 8.5 | 20.75 | 12.5 | 41.75 | 30 | 30.75 |
| Average hrs at night when observed | 1.06 | 2.66 | 2.83 | 2.59 | 4.17 | 2.61 | 2.00 | 2.05 |
| Dubious observations | 2 | 1 | 1 | 1 | 2 | 1 | 1 | 1 |
| Taken photos | 113 | 338 | 78 | 334 | 164 | 337 | 344 | 402 |
| Year | 1966 | 1967 | 1969 | 1973 | 1974 | 1975 | 1976 | |
| Observed, nights | 47 | 65 | 66 | 54 | 51 | 51 | 50 | |
| Certain MC sightings, times | 8 | 20 | 9 | 5 | 10 | 16 | 11 | |
| Certain MC sightings, hours | 18.25 | 39.25 | 15.5 | 12 | 25 | 36.75 | 24.75 | |
| Average hrs at night when observed | 2.28 | 1.96 | 1.72 | 2.4 | 2.5 | 2.3 | 2.25 | |
| Dubious observations | 3 | 0 | 1 | 2 | 1 | 3 | 4 | |
| Taken photos | 259 | 559 | 94 | 19 | 80 | 124 | 88 | |
| Year | 1977 | 1978 | 1979 | 1980 | 1981 | 1982 | 1983 | |
| Observed, nights | 58 | 41 | 40 | 37 | 28 | 35 | 42 | |
| Certain MC sightings, times | 16 | 8 | 5 | 5 | 7 | 8 | 10 | |
| Certain MC sightings, hours | 34 | 26.5 | 10.75 | 13.5 | 15.5 | 16.5 | 12.5 | |
| Average hrs at night when observed | 2.13 | 3.31 | 2.15 | 2.7 | 2.21 | 2.06 | 1.25 | |

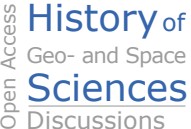

| Dubious observations | 2 | 2 | 1 | 0 | 1 | 0 | 0 | |
|---|---|---|---|---|---|---|---|---|
| Taken photos | 98 | 52 | 10 | 81 | 30 | 37 | 55 | |

**Table 5 Review of systematic observations of MC in Latvia: other observation points.**

| Year | **1957** | **1958** | **1959** | **1960** | **1961** | **1963** |
|---|---|---|---|---|---|---|
| Point of observation | Riga | Baldone | Baldone/Riga | Riga | Riga | Riga |
| Observed, nights | 86 | 94 | 76 | 45 | 70 | 30 |
| Certain MC sightings, times | 0 | 4 | 6 | 5 | 7 | 6 |
| Certain MC sightings, hours | 0 | 5 | 16.25 | 7.25 | 12.5 | 15.75 |
| Average hours at night when observed | | 1.25 | 2.71 | 1.45 | 1.79 | 2.63 |
| Dubious observations | 1 | 0 | 2 | 6 | 1 | 1 |
| Taken photos | 2 | 27 | 166 | 136 | 168 | 151 |
| Year | **1964** | **1965** | **1966** | **1967** | **1968** | **1969** |
| Point of observation | Riga | Riga | Riga | Riga | Riga | Riga |
| Observed, nights | 44 | 32 | 37 | 10 | 1 | 2 |
| Certain MC sightings, times | 8 | 7 | 6 | 8 | 1 | 2 |
| Certain MC sightings, hours | 15.24 | 15 | 12.75 | 13 | 2.75 | 2.25 |
| Average hours at night when observed | 1.91 | 2.14 | 2.13 | 1.63 | 2.75 | 2.25 |
| Dubious observations | 3 | 2 | 1 | 0 | 0 | 0 |
| Taken photos | 132 | 115 | 72 | 160 | 15 | 9 |

**Competing interests**

The author declare that they have no conflict of interest.

**Acknowledgments**

The author of the article expresses his deepest gratitude:

Dr. Paed. Ilgonis Vilks for his unselfish work in searching for the material of MM's observations in the still unsorted part of

the archive of the museum UL,

to the volunteer assistants Jānis Biķis and Jūlija Stepanova for their patient and persistent work in the digitization of the

observation journals.

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



Гришин Н.И., 1957. (составитель). Инструкция для наблюдений серебристых облаков. Изд. Академии Наук СССР,

Москва, 1957. 24с. (Instruction for observations of noctilucent clouds at IGY; in russian).

Дирикис М., 1957. Отчет Рижского полевого отряда о работе по серебристым облакам в 1957 г. Unpublished archive
document: Riga, 1957, 7p. (Report about Riga field expedition for observation of noctilucent clouds at 1957, in russian).

Дирикис М., Берзиньш Я., 1958. Отчет Рижского полевого отряда о работе по серебристым облакам в 1958 году.
Unpublished archive document: Riga, 1958, 11p. (Report about Riga field expedition for observation of noctilucent clouds at

1958, in russian).

Ромейко В.А., 1990. Руководство по проведению квалифицированных визуальных наблюдений серебристых облаков.
Москва, изд. МГДПиШ, 1990, 17с. (Guide for qualified visual observations of noctilucent clouds; in russian).

Ходоренко А., 2020. Небесное предупреждение. Над Европой появились необычные серебристые облака — что они
означают.        https://nv.ua/techno/popscience/serebristye-oblaka-stali-poyavlyatsya-vse-chashche-50098478.html.        Last

reviewed 2022.09.26. (in russian).