# Peer review of "Observations of mesospheric clouds in Latvia 1957-1983"

_History of Geo- and Space Sciences, 2024_

## Referee Comment (RC1)

Comments on

**Observations of mesospheric clouds in Latvia 1957–1983**

by Janis Kaulins

This manuscript contains details of visual and photographic NLC observations in Latvia from 1957-1983. It is very technical and lacks examples of observations.

The style is not up to international standards, but this can be fixed by later copy editing (generally performed by COPERNICUS staff).
The author uses mostly the abbreviation „MC" for the mesospheric clouds, but sometimes „NLC", and even „MM". It is recommended to use either NLC or PMC (polar mesospheric clouds) throughout the manuscript which are the generally accepted abbreviations by the scientific community.
The manuscript contains many tables, but no examples of a photographic observation.
From line 50 on the author mentions some work performed with the data, Even if these works are no longer available it would be interesting for the reader to know about their content and results. This manuscript contains otherwise no scientific results.
The author should also explain, why the observations stopped in 1983 and why they are not continued.
As the author pointed out, the archived material may be useful for further studies on NLC. Therefore this manuscript can be published in HGSS after considerable revision.

**Detailed criticism and suggestions:**
line 30: I suggest to cite „Hodorenko, 2020" as „Ходоренко, 2020" in order to locate it easier in the ref. list
line 42: what means LAB? Should it read LAS?
line 46: same as line 30 -  cite it as "Гришин, 1957".
line 48: same as line 30 -  „Ромейко, 1990"
line 48: „Romeyko et al." is written „Romejko et al." in the reference list
line 52: since Mukin's article is no longer available, the author should include a few words about ist content and results
line 58: cite „Gadsden **and** Taylor, and „Olivieiro **and** Thomas"
line 71: same as line 30 - „Дирикис М., 1957". and „Дирикис and Берзиньш Я., 1958"
Table 1: it should be explained what the 4th column „AMSL.,m" means
lines 130-141: the EXEL tables contained in the e-source repository (https://dspace.lu.lv/dspace/handle/7/61099) are not very useful to interested international scientists, because the columns and explanations are only in Latvian. If the author wants to provide this material for the international scientific community, English translations should be added. This should be mentioned in the manuscript.
Fig. 4: This photo is hardly readable. The authors should provide a better reproduction and also explain the meaning of the columns. The author may consider to omit this figure and instead include a few lines from the archived EXEL table as example.

line 160: cite „Bronshtein **and** Grishin"
line 170: what means „AVRM"?
line 187: are the images made in color?
Fig. 8: This figure does not contain useful information. Instead, examples of photos should be included. Perhaps a  very good observation with pronounced structure and a „dubious" one (as listed in Table 4).
line 206: can the author give an estimate, when the digitizes images will be available?
line 208: cite „Eglītis **and** Eglīte, 2016"
section 2.5.2: this section can be shortened, it would be sufficient to give the details of the used Russian cameras (incl. Fig 6) with the given Russian reference. What means FED-3 and Viliya?
lines 217 and 221: same as line 30 -  „Абрамов"
line 261: „MC height information should be taken…" Does this mean, the archived date do not contain height information?
line 265: „… shows an apparent inverse…". Can a reference be given for this result?
line 307: I suggest to insert a subheading here: „References in Russian"